# Inflammatory Blood Markers as Prognostic and Predictive Factors in Early Breast Cancer Patients Receiving Neoadjuvant Chemotherapy

**DOI:** 10.3390/cancers12092666

**Published:** 2020-09-18

**Authors:** Ileana Corbeau, Simon Thezenas, Aurelie Maran-Gonzalez, Pierre-Emmanuel Colombo, William Jacot, Severine Guiu

**Affiliations:** 1Department of Medical Oncology, Institut du Cancer de Montpellier (ICM), Parc Euromedecine, 208 avenue des apothicaires, 34 298 Montpellier, France; William.Jacot@icm.unicancer.fr (W.J.); severine.guiu@icm.unicancer.fr (S.G.); 2Institut de Recherche en Cancerologie de Montpellier, INSERM U1194, University of Montpellier, ICM Parc Euromedecine, 208 Avenue des Apothicaires, 34 298 Montpellier, France; Pierre-Emmanuel.Colombo@icm.unicancer.fr; 3Department of Biostatistics, Cancer Institute of Montpellier (ICM), University of Montpellier, 34 298 Montpellier, France; simon.thezenas@icm.unicancer.fr; 4Department of Anatomopathology, ICM Parc Euromedecine, 208 Avenue des Apothicaires, 34 298 Montpellier, France; aurelie.maran-gonzalez@icm.unicancer.fr; 5Department of Oncologic Surgery, ICM Parc Euromedecine, 208 Avenue des Apothicaires, 34 298 Montpellier, France

**Keywords:** breast cancer, neoadjuvant chemotherapy, inflammatory blood marker, predictive factor, prognostic factor, pathological complete response

## Abstract

**Simple Summary:**

Predictive and prognostic factors are necessary to evaluate the future of women with early breast cancer. Inflammatory blood markers such as neutrophil to lymphocytes ratio and platelet to lymphocytes ratio have been reported to be a predictive factor for pathological complete response and a prognostic factor in breast cancer, with conflicting results. Here we evaluate these inflammatory blood markers in patients with early breast cancer receiving neo adjuvant chemotherapy since neo adjuvant treatment is more and more developed in early breast cancer.

**Abstract:**

Background: Inflammatory blood markers, such as neutrophil to lymphocyte ratio (NLR) and platelet to lymphocyte ratio (PLR), have been reported as putative prognostic factors for survival and predictive factors for pathological complete response and toxicity in cancers, however with conflicting results. Methods: We retrospectively analyzed data of 280 patients with early breast cancer receiving neo-adjuvant chemotherapy between 2005 and 2013 in our center. Neutrophil count, lymphocyte count and platelet count before treatment were collected as well as data on pathological complete response, toxicity, recurrence and survival. Results: In multivariate analysis, high PLR was an independent prognostic factor for relapse-free survival (hazard ratio [HR] = 1.91; 95%CI = 1.15–3.16; *p* = 0.012) and for shorter overall survival (HR = 1.83; 95%CI = 1.03–3.24; *p* = 0.039). NLR was an independent predictive factor for febrile neutropenia (HR = 0.28; 95%CI = 0.13–0.58; *p* = 0.001). In triple negative breast cancer molecular subtype, low white blood cell count (<6.75 G/L) was predictive for a higher pathological complete response rate (odds ratio [OR] = 0.29; 95%CI = 0.14–0.61; *p* < 0.01). Conclusion: In the present study, PLR was found as an independent prognostic factor for survival, while NLR was an independent predictive factor for febrile neutropenia.

## 1. Introduction

Breast cancer is the most common female cancer (with over two million new cases worldwide in 2018) and the leading cause of cancer-related death in women [1]. Although more and more patients undergo breast cancer treatment, the mortality rate has decreased in developed countries due to improvement in treatments and early detection. Today’s challenge is not only to cure these patients but also to reduce treatment-related toxicity. Neo-adjuvant chemotherapy has increasingly become used in early breast cancer treatment, especially for patients with HER2 positive or triple negative breast cancer. It allows the evaluation of the pathological response to chemotherapy, which has been shown to be correlated with survival [2], since patients achieving pathological complete response are showing improved overall survival (OS) and relapse-free survival (RFS) [3,4]. Inflammatory blood markers (IBM) have emerged as potential prognostic factors for survival in different types of cancers including breast cancer, as well as predictive factors for histological response after neo-adjuvant chemotherapy [5,6,7,8,9,10,11]. IBM include leucocyte count, lymphocyte count, neutrophil count, and ratios such as platelet to lymphocyte ratio (PLR) or neutrophil to lymphocyte ratio (NLR), the last one being the most evaluated. Several studies have been published on IBM as predictive for pathological complete response in early breast cancer patients receiving neo-adjuvant chemotherapy, however with conflicting results [12,13]. In the present study, we evaluated if IBM are prognostic for relapse-free survival (RFS) and/or overall survival (OS) in a large series of homogeneously treated early breast cancer patients receiving sequential anthracycline–taxane neo-adjuvant chemotherapy, and if these IBM are predictive for pathological complete response in the same population of patients. We also evaluated if there was an association between IBM and neo-adjuvant chemotherapy-related toxicity.

## 2. Results

### 2.1. Patients’ Characteristics

Two hundred and eighty patients were included in the present study. Patients’ characteristics are described in Table 1. The median age at diagnosis was 50.3 years old and 36.8% of our patients were menopaused at diagnosis. Clinical stage was mainly cT2 (58.9%) and half of the patients (51.8%) had clinical lymph node invasion. Majority of patients were affected by an invasive ductal carcinoma (84.0%) and histological grade three disease (52.2%). One hundred and twenty-seven patients (45.4%) had hormone receptor positive/HER2 negative breast cancer, 41 patients (14.6%) had hormone receptor positive/HER2 positive breast cancer, 40 patients (14.3%) had hormone receptor negative/HER2 positive breast cancer and 72 patients (25.7%) had triple negative breast cancer. Eleven patients (6.3%) were known to be *BRCA* 1 or 2 deleterious mutation carriers. All patients were selected as fit for neo-adjuvant chemotherapy and no major comorbidity was reported in our population.

Before any treatment, median white blood cell count was 6.8 G/L, median lymphocyte count was 1.9 G/L, median neutrophil count was 3.9 G/L, median NLR was 2.0 and median PLR was 132.3. Patients’ clinical and biological characteristics are summarized in Table 1.

Ninety-one percent of the patients completed the planned neo-adjuvant chemotherapy sequence. Main causes of interruption were toxicity (3.9%, *n* = 11 patients) and progression (1.4%, *n* = 4). Seventy percent of the patients had breast-conserving surgery (*n* = 198) and 98.9% had lymph node surgery (*n* = 276). Ninety-seven percent of patients received adjuvant radiation therapy (*n* = 272), 55.7% of patients received adjuvant hormone therapy (*n* = 156) (as a reminder 60% of patients were affected by a hormone receptor positive breast cancer) and 27.9% received neoadjuvant/adjuvant trastuzumab (*n* = 78) (28.9% had HER2 positive breast cancer).

Cut-offs for NLR and PLR calculated using receiver operating characteristic (ROC) curves gave 2.4 and 131.1 cut-off values, with AUC-ROC values of 0.55 and 0.58, respectively. These cut-offs were not associated with any significant predictive or prognostic value (except for PLR and RFS but only in univariate analysis) (data not shown). Therefore, we used previously published cut-offs for NLR and PLR at 2 and 150, respectively, according to the literature [14,15,16,17,18,19].

### 2.2. Predictive Factors for Pathological Complete Response

Seventy-four patients (26.4%) reached pathological complete response. In univariate analysis (Table 2), histological grade (*p* = 0.001), mitotic index (*p* = 0.006), hormone receptor expression (*p* < 0.001) and molecular subtype (*p* < 0.001) were correlated with pathological complete response. None of the IBM were significantly correlated with pathological complete response. In multivariate analysis (Table 2), nodal status (odds ratio (OR) = 0.43; 95%CI = 0.23–0.82; *p* = 0.01), histological grade (grade 2: OR = 0.13; 95%CI = 0.07–0.27; *p* < 0.01 and grade 3: OR = 0.28; 95%CI = 0.12–0.63; *p* < 0.01 versus grade 1) and molecular subtype (hormone receptor positive/HER2 positive: OR = 7.98; 95%CI = 0.12–0.63; *p* < 0.01; hormone receptor negative/HER2 positive: OR = 11.94; 95%CI = 4.76–29.91; *p* > 0.01 and hormone receptor negative/HER2 negative: OR = 4.65; 95%CI = 1.98–10.89; *p* < 0.01; compared to hormone receptor positive/HER2 positive) were independent predictive factors of pathological complete response.

### 2.3. Prognostic Factors for RFS

With a median follow-up of 6.7 years (range 0.2 to 11.3) at the date of data cleaning (March 2019), 65 patients (23.2%) had relapsed, including 51 patients with metastatic relapse and 14 patients with locoregional relapse.

Higher cT stage (*p* = 0.005), inflammatory breast cancer (*p* = 0.003), hormone receptor negative breast cancer (*p* = 0.003), triple negative breast cancer (*p* = 0.001), not reaching pathological complete response (*p* = 0.009), low neutrophil count (<1.5 G/L) (*p* = 0.007), and high PLR (≥150) (*p* = 0.044) were all correlated with a higher risk of relapse in univariate analysis (Table 3).

In multivariate analysis, high PLR was still an independent prognostic factor (HR = 1.91; 95%CI = 1.15–3.16; *p* = 0.012) as well as low neutrophil count (<1.5 G/L) (HR = 0.06; 95%CI = 0.03–0.12; *p* < 0.001). Other independent prognostic factors for RFS were high tumor size (HR = 2.07; 95%CI = 1.26–3.39; *p* = 0.004), triple negative breast cancer subtype (HR = 4.11; 95%CI = 2.28–7.42; *p* < 0.001) and pathological complete response achievement (HR = 0.27; 95%CI = 0.12–0.56; *p* = 0.001) (Table 3).

### 2.4. Prognostic Factors for OS

At the end of follow-up, 51 patients (18.2%) had died, including 40 breast cancer-related deaths. Other deaths were due to a second cancer for six patients, chemotherapy-related toxicity (cardiac failure) for one patient, intercurrent disease for one patient and unknown for three patients.

In univariate analysis (Table 4), the following factors were correlated with a poor survival: higher tumor size (*p* = 0.006), inflammatory breast cancer (*p* = 0.014), hormone receptor negative breast cancer (*p* = 0.001), triple negative breast cancer (*p* < 0.001), not reaching pathological complete response (*p* = 0.041) and platelet count ≥ 264 G/L (*p* = 0.049).

In multivariate analysis (Table 4), high PLR and low neutrophil count (<1.5 G/L) were associated with shorter OS (HR = 1.83; 95%CI = 1.03–3.24; p = 0.039 and HR = 0.12; 95%CI = 0.06–0.24; *p* < 0.001, respectively). Other independent prognostic factors for OS were high tumor size (HR = 2.43; 95%CI = 1.37–4.30; *p* = 0.002), triple negative breast cancer subtype (HR = 6.59; 95%CI = 3.34–13.01; *p* < 0.001) and pathological complete response achievement (HR = 0.29; 95%CI = 0.13–0.66; *p* = 0.003).

### 2.5. Sub Group Analyses

Platelet count, neutrophil count, lymphocyte count, PLR or NLR were not predictive factors for pathological complete response in any of the molecular subtypes. Low white blood cell count (<6.75 G/L) was an independent predictive factor for pathological complete response in patients with triple negative breast cancer (OR = 0.29; 95%CI = 0.14–0.61; *p* < 0.01), but not for other molecular subtypes. Data for survival were not mature enough to highlight prognostic factors according to the molecular subtype.

### 2.6. Predictive Factors for Febrile Neutropenia 

Thirty-eight patients (13.6%) developed febrile neutropenia during neo-adjuvant chemotherapy. Post-menopausal status (*p* = 0.005), negative hormone receptor (*p* = 0.003), low platelet count (<264 G/L) (*p* = 0.015), low white blood cell count (<6.75 G/L) (*p* = 0.002) and low NLR (*p* = 0.001) were predictive for febrile neutropenia in univariate analysis (Table 5). Molecular subgroup was predictive for febrile neutropenia since 34.9% of positive hormone receptor-HER2 negative patients, 32.6% of negative hormone receptor-HER2 negative patients and 27.9% of negative hormone receptor-HER2 positive patients developed febrile neutropenia. Only 4.7% of patients with positive hormone receptor and positive HER2 status developed this toxicity (*p* = 0.007).

In multivariate analysis (Table 5), low NLR was independently predictive for febrile neutropenia (OR = 0.28; 95%CI = 0.13–0.58; *p* = 0.001) as well as low platelet count (<264 G/L) and low white blood cell count (<2.75 G/L) (OR = 0.40; 95%CI = 0.21–0.75; *p* = 0.004 and OR = 0.35; 95%CI = 0.17–0.72; *p* = 0.004, respectively). Clinical tumor size was also a predictive factor for febrile neutropenia (OR = 0.43; 95%CI = 0.20–0.91; *p* = 0.028). Patients with positive hormone receptor and positive HER2 status developed less febrile neutropenia than patients with positive hormone receptor and negative HER2 status (OR = 0.21; 95%CI = 0.005–0.93; *p* = 0.039). Other molecular subtypes were not independently associated with febrile neutropenia.

We also calculated a cut-off for NLR to be predictive for febrile neutropenia, this cut-off point was set at 4.125, meaning that patients with an initial NLR higher than 4.125 had less risk of febrile neutropenia (AUC = 0.52; *p* = 0.308). Seventeen patients (6.1%) had NLR ≥ 4.125 and the risk of febrile neutropenia for this group was 1.4%. Two hundred and sixty-three patients (93.1%) had NLR lower than 4.125 and the risk of febrile neutropenia for this group was 13.9%.

## 3. Discussion

This is the second largest cohort of patients published regarding IBM as predictive factors (after Graziano et al.’s study [20]) and the largest population with reported multivariate results regarding IBM as prognostic factors in patients with localized breast cancer homogeneously receiving neo-adjuvant chemotherapy. Our population is representative for patients receiving neo-adjuvant chemotherapy since most of our patients were affected by cT2 stage (or higher), node-positive breast cancer. We chose to include the four molecular subtypes in order to have a comprehensive evaluation of the correlations between IBM and neo-adjuvant chemotherapy in breast cancer patients.

Most of the studies evaluating IBM as prognostic factors studied Asian-based populations [9,14,16,18,19,21,22,23,24,25,26,27,28,29,30,31,32,33], where use of anti-cancer drugs, particularly trastuzumab, is inconsistently reported, inducing heterogeneity in survival expectations. This demographic variable could be associated with specific molecular and pharmacological features that can induce differences in toxicity and efficacy profiles. We report here a large population of Western breast cancer patients receiving sequential anthracycline and taxane-based chemotherapy, with access to anti-HER2 therapies.

Another strength of our series is the long clinical follow-up, with a median duration of more than six years, a substantially longer follow-up than most similar studies, allowing a relevant and mature evaluation of survival data.

Finally, we are the first ones to evaluate the relation between IBM and toxicity in a population of early breast cancer patients receiving neo-adjuvant chemotherapy, and we found that low NLR was an independent predictive factor for febrile neutropenia.

One limitation comes from the monocentric, retrospective nature of our studied population. However, the clinicopathological characteristics of our population and our results are consistent with previously published studies evaluating IBM. In a closely-related population of breast cancer patients receiving adjuvant treatment, Ramos Esquivel et al. [34], Takeuchi et al. [21] and Cho et al. [28] also showed that PLR was a prognostic factor for disease-free survival (DFS) and OS in univariate and multivariate analyses. In a population of patients with metastatic triple negative breast cancer, Vernieri et al. [35], found that PLR was statistically significantly associated with progression-free survival. Only one study (written by Losada et al. [36]) evaluating PLR as prognostic factor in patients receiving neo-adjuvant chemotherapy failed to show evidence of a relation between PLR and DFS. Additionally, their population was different to ours since they included only elderly patients (≥65 years old).

Results on NLR and survival are more controverted. In populations receiving neo-adjuvant chemotherapy, two studies (Suppan et al. [37], and Marin Hernandez et al. [38]) found, similarly to us, that NLR was not prognostic for DFS and/or OS, whereas two other studies (Chen et al. [14], and Koh et al. [23]) found that NLR was associated with survival. These conflicting results could be due to a difference in the studied population since both Chen et al. and Koh et al. had fewer or no triple negative breast cancer patients in their studied population, while Suppan et al., Marin Hernandez et al. and our study included over 20% of triple negative breast cancer patients. NLR prognostic value appears more clearly validated in the population of patients receiving adjuvant chemotherapy, since many studies have demonstrated a significant correlation in this setting [16,19,24,26,29,31,33,39,40,41,42,43].

In our population, we found no significant association between NLR or PLR with pathological complete response, concordantly with previous reports [17,20,22,36,37] evaluating a similar population of breast cancer patients receiving neo-adjuvant chemotherapy. Only one study (Chae et al. [18]) found that NLR was predictive for pathological complete response; however, they included only Chinese patients with triple negative breast cancer, differing strikingly from our population. Cuello Lopez et al. [44] reported, in univariate analysis, that patients with low PLR had higher pathological complete response rates, but no multivariate data were available in this publication.

As previously said, we are the first study, to our knowledge, to analyze data on the correlations between IBM and chemotherapy-related toxicity in breast cancer patients receiving neo-adjuvant chemotherapy. Evaluation of IBM, accessible through a simple blood analysis, could appear as an interesting, inexpensive tool to individualize one’s hematological risk under neo-adjuvant chemotherapy. Considering NLR as a predictive factor for febrile neutropenia could help to decide, for example, which patients should benefit from granulocyte colony stimulating factors in intermediate risk situations [45]. The only data available on NLR and toxicity in this situation were published by Yamanouchi et al. [46] and they found no significant correlation. However, the population was limited (67 metastatic breast cancer patients), possibly explaining the lack of statistical significance. Two authors evaluated lymphopenia as a predictive factor for toxicity. Ray Coquard et al. conducted two studies [47,48] showing that lymphocyte count at day one of chemotherapy < 0.7 G/L was an independent prognostic factor for early death after chemotherapy and for febrile neutropenia, in a large cohort of patients (including breast cancer patients). Choi et al. [49], on the contrary, could not establish a relation between day one lymphopenia and febrile neutropenia in a cohort of patients (including 13% of breast cancer patients) receiving their first course of chemotherapy.

Another limitation of our work is our inability to robustly evaluate if IBM could be predictive for non-hematological toxicities, due to a small number of these events in our retrospective series. Dedicated studies on cardiac, neurological, or hepatic toxicities remains necessary to draw conclusions regarding possible associations between IBM and these toxicities.

NLR has also been studied in other cancers, and more specifically in patients receiving neo-adjuvant chemotherapy. Chen et al. [50] found a relation between NLR and DFS/OS in univariate but not in multivariate analysis in 91 patients receiving neo-adjuvant chemotherapy for stomach cancer. Shen et al. [51] did not find any correlation between NLR and survival in 202 patients receiving neoadjuvant chemo-radiotherapy for locally advanced rectal cancer. Buisan et al. [52] found that NLR was prognostic for survival in 50 patients with squamous cell featuring bladder cancer receiving neo-adjuvant chemotherapy. Li et al. [13] published a large meta-analysis on 6243 patients affected by different kinds of cancer (including 1507 breast cancer) receiving neo-adjuvant chemotherapy. He showed that high NLR was associated with a lower pathological complete response rate, low NLR was associated with better OS, cancer-specific survival, DFS and recurrent-free survival. Whether the discrepancies between our work and this meta-analysis are linked to statistical power or heterogeneity of the prognostic value of NLR between tumor types needs to be further evaluated.

## 4. Materials and Methods

### 4.1. Patients

Patients treated with neo-adjuvant chemotherapy for early breast cancer in our institution between January 2005 and September 2013 were included in a clinico-pathological database [53,54,55]. The database follow-up has been updated and completed until March 2019. Patients included in our study presented with unilateral, unifocal, non-metastatic early breast cancer confirmed by core needle biopsy and had received sequential anthracycline–taxane neo-adjuvant chemotherapy. Initial blood samples were collected before starting any treatment. Patients without available initial blood sample analysis were excluded from the present study.

### 4.2. Pathological Assessments

Hormone receptor positive breast cancer was defined as tumors with ≥ 10% of estrogen and/or progesterone receptors immunohistochemical (IHC) expression. HER2 positive breast cancer was defined as tumors with HER2+++ on IHC staining or tumors with IHC HER2++ expression and average HER2 gene copy numbers ≥ 6 at cases with a HER2/CEP17 ratio of less than 2 or a HER2/CEP17 ratio of 2 or more, independently of the average gene copy number after fluorescent in situ hybridization (FISH) [56].

### 4.3. Treatments

All patients received three to four cycles of anthracycline-based chemotherapy, followed by three to four cycles of taxane (weekly paclitaxel or docetaxel every 3 weeks). Patients with HER2 positive tumors received neoadjuvant trastuzumab concomitantly with taxane, followed by adjuvant trastuzumab for a total of 12 months. Patients with hormone receptor positive received adjuvant hormone therapy for 5 years. All patients received adjuvant radiation therapy according to ESMO (European Society of Medical Oncology) guidelines [57].

### 4.4. Endpoints

Pathological complete response was defined as the absence of invasive cancer in the breast and axillary nodes, irrespective of the presence of in situ ductal carcinoma (ypT0/is ypN0). RFS was defined as the time from the date of biopsy to the date of documented relapse (local, loco-regional or distant recurrence) or death from progression. OS was defined as the time from the date of biopsy to the date of documented death from any cause.

For the toxicity endpoint, febrile neutropenia was defined as temperature ≥38.5 °C and neutrophil count ≤ 0.5 G/L [58].

As no standard definition of relevant IBM is available, different cut-offs for NLR and PLR were tested. First, a cut-off was set using receiver operating characteristic (ROC) curves, a second one was based using the cut-offs previously published [14,15,16,21,39,59]. We also calculated a cut off for NLR to be statistically associated with a risk of febrile neutropenia.

### 4.5. Ethical Statement

This study was reviewed and approved by the Montpellier Cancer Institute Institutional Review Board (ID number ICM-CORT-2019-41). Due to the non-interventional, based on clinical routine data, nature of the study, the Institutional Review Board waived the need for written, informed consent.

### 4.6. Statistical Analysis

Descriptive analyses were performed using median and range for continuous parameters, frequency and percentage for categorical variables.

Baseline characteristics of included patients were compared by Kruskal–Wallis tests or Wilcoxon for continuous variables, or chi2 or Fisher’s exact test for categorical variables.

The median follow-up was calculated using the reverse Kaplan–Meier method with its confidence interval (CI) of 95%. All event-free survival (RFS, OS) was estimated using the Kaplan–Meier method, and then described using medians and rates with their associated 95% CI.

Survival curves were drawn, and the log-rank test was performed to assess differences between groups.

Moreover, multivariate analyses were carried out using logistic regressions or Cox’s proportional hazards regressions, with a stepwise selection procedure, to investigate known predictive or prognostics factors, respectively.

Odds ratios (OR) and hazard ratios (HR), with their 95% Cis, are presented to display reductions in risk.

ROC curves were used to discriminate the optimal threshold for individual markers (NLR/PLR), calculated to maximize the Youden’s index (gold standard: relapse).

All *p* values reported are two-sided, and the significance level was set at 5% (*p* < 0.05). Statistical analysis was performed using the STATA 13.1 software (Stata Corporation, College Station, TX, USA).

## 5. Conclusions

We report here the correlations between IBM and clinical endpoints in a homogeneous population of 280 breast cancer patients treated with neo-adjuvant chemotherapy in a single institution. We show that PLR is an independent prognostic factor for RFS and OS and that NLR is an independent predictive factor for febrile neutropenia. None of the IBM were predictive for pathological complete response in our overall population. When focusing on specific molecular subtypes, white blood cell count was an independent predictive factor for pathological complete response in triple negative breast cancer. In addition, we found a highly significant association between pathological complete response and survival, pathological complete response being prognostic both for RFS and OS in univariate and multivariate analyses in our study.

IBM are an easy and efficient tool to evaluate patient’s prognosis and the risk of febrile neutropenia. Additional studies evaluating larger populations, as well dedicated prospective studies, are needed in order to evaluate if IBM could also be reliably predictive for pathological complete response.

## Figures and Tables

**Table 1 cancers-12-02666-t001:** Patients’ characteristics.

Characteristics	No (%)
Age (median)	50.3 (25.3–76.6)
Menopausal status	
Yes	103 (36.8)
No	177 (63.2)
T stage (before NACT)	
T0	2 (0.7)
T1	22 (7.9)
T2	165 (58.9)
T3	47 (16.8)
T4	43 (15.4)
Missing	1 (0.3)
Node status (before NACT)	
N-	129 (46.1)
N+	145 (51.8)
Missing	6 (2.1)
Inflammatory BC	
Yes	39 (14.2)
No	241 (85.8)
Histology	
Ductal	235 (84)
Lobular	27 (9.6)
Other	18 (6.4)
Histological grade	
1	5 (1.8)
2	123 (43.9)
3	146 (52.2)
Missing	6 (2.1)
Differentiation (*N* = 215)	
1	1 (0.5)
2	21 (9.8)
3	193 (89.7)
Nucleus (*N* = 216)	
1	3 (1.4)
2	77 (35.6)
3	136 (63)
Mitosis (*N* = 216)	
1	70 (32.4)
2	70 (32.4)
3	76 (35.2)
HER2 status	
Positive	81 (28.9)
Negative	199 (71.1)
HR status	
Positive	168 (60)
Negative	112 (40)
Molecular subtype	
HR+/HER2₋	127 (45.4)
HR+/HER2+	41 (14.6)
HR₋/HER2+	40 (14.3)
HR₋/HER2₋	72 (25.7)
Baseline biology: median (range)	
Hemoglobin (g/dl)	13.6 (8.0–15.4)
Platelets (G/L)	264 (124–452)
White blood cell count (G/L)	6.8 (1.5–30)
Lymphocyte count (G/L)	1.9 (0.6–26)
PNN (G/L)	3.9 (1.4–11.6)
NLR	2.0 (0.1–17.3)
PLR	132.3 (6.9–514.9)

NACT: neo adjuvant chemotherapy, BC: breast cancer, HR: hormone receptor, PNN: neutrophils, NLR: neutrophil to lymphocyte ratio, PLR: neutrophil to lymphocyte ratio.

**Table 2 cancers-12-02666-t002:** Univariate and multivariate logistic regression of predictive factors for pathological complete response.

PCR	Univariate				Multivariate	
Variable	No	Yes	Total	*p*	Odd Ratio (95%CI)	*p*
Age (year)				0.278		
<50.25	99 (48.1%)	41 (55.4%)	140 (50%)			
≥50.25	107 (51.9%)	33 (44.6%)	140 (50%)			
cT				0.895		
T 0/2	140 (68.0%)	49 (67.1%)	189 (67.7%)			
T 3/4	66 (32.0%)	24 (32.9%)	90 (32.3%)			
cN				0.093		
Negative	88 (44.0%)	41 (55.4%)	129 (47.1%)		1	0.010
Positive	112 (56.0%)	33 (44.6%)	145 (52.9%)		0.43 (0.23–0.82)	
Inflammatory BC				0.967		
No	174 (85.7%)	61 (85.9%)	235 (85.8%)			
Yes	29 (14.3%)	10 (14.1%)	39 (14.2%)			
Histological grade				0.001		
1					1	
2	106 (52.5%)	22 (30.6%)	128 (46.7%)		0.13 (0.07–0.27)	<0.001
3	96 (47.5%)	50 (69.4%)	146 (53.3%)		0.28 (0.12–0.63)	0.002
Histologic type				0.233		
Ductal	168 (81.5%)	67 (90.5%)	235 (83.9%)			
Lobular	23 (11.2%)	4 (5.4%)	27 (9.7%)			
Others	15 (7.3%)	3 (4.1%)	18 (6.4%)			
Hormone receptor				<0.001		
Negative	65 (31.6%)	47 (63.5%)	112 (40.0%)			
Positive	141 (68.4%)	27 (36.5%)	168 (60.0%)			
Molecular subtype				<0.001		
HR+HER2₋	116 (56.3%)	11 (14.9%)	127 (45.4%)		1	
HR+HER2+	25 (12.1%)	16 (21.6%)	41 (14.6%)		7.98 (3.13–20.34)	<0.001
HR₋HER2+	18 (8.8%)	22 (29.7%)	40 (14.3%)		11.94 (4.76–29.91)	<0.001
HR₋HER2₋	47 (22.8%)	25 (33.8%)	72 (25.7%)		4.65 (1.98–10.89)	<0.001
Platelet count				0.068		
<264 G/L	97 (47.1%)	44 (59.5%)	141 (50.4%)			
≥264 G/L	109 (52.9%)	30 (40.5%)	139 (49.6%)			
White blood cell count				0.058		
<6.75 G/L	96 (46.6%)	44 (59.5%)	140 (50.0%)			
≥6.75 G/L	110 (53.4%)	30 (40.5%)	140 (50.0%)			
Neutrophil count				1.000		
<1.5 G/L	1 (0.5%)	0 (0.0%)	1 (0.4%)			
≥1.5 G/L	205 (99.5%)	74 (100.0%)	279 (99.6%)			
Lymphocyte count				0.685		
<1 G/L	7 (3.4%)	1 (1.4%)	8 (2.9%)			
≥1 G/L	199 (96.6%)	73 (98.6%)	272 (97.1%)			
NLR				0.550		
<2	103 (50.0%)	34 (45.9%)	137 (48.9%)			
≥2	103 (50.0%)	40 (54.1%)	143 (51.1%)			
PLR				0.617		
<150	132 (64.1%)	45 (60.8%)	177 (63.2%)			
≥150	74 (35.9%)	29 (39.2%)	103 (36.8%)			

BC: breast cancer, cT: clinical tumor size, cN: clinical node involvement, HR: hormone receptor, NLR: neutrophil to lymphocyte ratio, PLR: platelet to lymphocyte ratio.

**Table 3 cancers-12-02666-t003:** Univariate and multivariate logistic regression of prognostic factors for recurrence-free survival (RFS).

RFS	Univariate		Multivariate	
Variable	Hazard Ratio (95%CI)	*p*	Hazard Ratio (95%CI)	*p*
Age (year)				
<50.25	1			
≥50.25	1.50 (0.91–2.46)	0.111		
cT				
T 0/2	1		1	
T 3/4	2.01 (1.23–3.26)	0.005	2.07 (1.26–3.39)	0.004
cN				
Negative	1			
Positive	1.61 (0.97–2.69)	0.066		
Inflammatory BC				
No	1			
Yes	2.34 (1.33–4.12)	0.003		
Histological grade				
1/2	1		1	
3	1.12 (0.68–1.84)	0.657	0.84 (0.47–1.51)	0.65
Histologic type				
Ductal	1			
Lobular	0.93 (0.40–2.16)	0.866		
Others	1.84 (0.84–4.06)	0.129		
HR				
Negative	1			
Positive	0.48 (0.29–0.78)	0.003		
Molecular subtype				
HR+HER2₋	1		1	
HR+HER2+	1.135 (0.505–2.551)	0.759	1.55 (0.73–3.33)	0.253
HR₋HER2+	1.378 (0.634–2.993)	0.418	1.89 (0.86–4.14)	0.111
HR₋HER2₋	2.669 (1.511–4.714)	0.001	4.11 (2.28–7.42)	<0.001
PCR				
No	1		1	
Yes	0.37 (0.18–0.78)	0.009	0.27 (0.12–0.56)	0.001
Platelet count				
<264 G/L	1			
≥264 G/L	1.73 (1.05–2.84)	0.031		
White blood cell count				
<6.75 G/L	1			
≥6.75 G/L	1.14 (0.70–1.86)	0.593		
Neutrophil count				
<1.5 G/L	1		1	
≥1.5 G/L	0.06 (0.01–0.48)	0.007	0.06 (0.03–0.12)	<0.001
Lymphocyte count				
<1 G/L	1			
≥1 G/L	0.89 (0.22–3.64)	0.871		
NLR				
<2	1			
≥2	1.02 (0.63–1.66)	0.930		
PLR				
<150	1		1	
≥150	1.65 (1.01–2.69)	0.044	1.91 (1.15–3.16)	0.012

cT: clinical tumor size, cN: clinical node involvement, BC: breast cancer, HR: hormone receptor, PCR: pathological complete response, NLR: neutrophil to lymphocyte ratio, PLR: platelet to lymphocyte ratio.

**Table 4 cancers-12-02666-t004:** Univariate and multivariate logistic regression of prognostic factors for overall survival (OS).

OS	Univariate		Multivariate	
Variable	Hazard Ratio (95%CI)	*p*	Hazard Ratio (IC95%CI)	*p*
Age (year)				
<50.25	1			
≥50.25	1.60 (0.91–2.83)	0.103		
cT				
T 0/2	1		1	
T 3/4	2.15 (1.24–3.73)	0.006	2.43 (1.37–4.30)	0.002
cN				
Negative	1			
Positive	1.42 (0.80–2.51)	0.232		
Inflammatory BC				
No	1			
Yes	2.21 (1.17–4.16)	0.014		
Histological grade				
1/2	1			
3	1.00 (0.57–1.76)	0.993		
Histologic type				
Ductal	1			
Lobular	0.76 (0.27–2.12)	0.604		
Others	1.20 (0.43–3.33)	0.732		
HR				
Negative	1			
Positive	0.37 (0.21–0.65)	0.001		
Molecular subtype				
HR+HER2₋	1		1	
HR+HER2+	1.33 (0.15–3.47)	0.555	1.77 (0.72–4.34)	0.211
HR₋HER2+	1.15 (0.41–3.20)	0.785	1.44 (0.50–4.13)	0.501
HR₋HER2₋	4.15 (2.16–7.96)	<0.001	6.59 (3.34–13.01)	<0.001
PCR				
No	1		1	
Yes	0.43 (0.20–0.97)	0.041	0.29 (0.13–0.66)	0.003
Platelet count				
<264 G/L	1			
≥264 G/L	1.76 (1.00–3.09)	0.049		
White blood cell count				
<6.75 G/L	1			
≥6.75 G/L	1.39 (0.80–2.41)	0.248		
Neutrophil count				
<1.5 G/L	1		1	
≥1.5 G/L	0.16 (0.02–1.17)	0.071	0.12 (0.06–0.24)	<0.001
Lymphocyte count				
<1 G/L	1			
≥1 G/L	0.70 (0.71–2.89)	0.624		
NLR				
<2	1			
≥2	0.96 (0.55–1.66)	0.890		
PLR				
<150	1		1	
≥150	1.49 (0.85–2.60)	0.160	1.83 (1.03–3.24)	0.039

cT: clinical tumor size, cN: clinical node involvement, BC: breast cancer, HR: hormone receptor, PCR: pathological complete response, NLR: neutrophil to lymphocyte ratio, PLR: platelet to lymphocyte ratio.

**Table 5 cancers-12-02666-t005:** Multivariate logistic regression of predictive factors for febrile neutropenia.

Febrile Neutropenia	Univariate				Multivariate	
Variable	No	Yes	Total	*p*	Odd Ratio (95%CI)	*p*
Age (year)			1			
<50.25	124 (52.3%)	16 (37.2%)	40 (50.0%)			
≥50.25	113 (47.7%)	27 (62.8%)	140 (50.0%)	0.068		
cT						
T 0/2	156 (66.1%)	33 (76.7%)	189 (67.7%)		1	
T 3/4	80 (33.9%)	10 (23.3%)	90 (32.3%)	0.170	0.43 (0.20–0.91)	0.028
cN						
Negative	107 (46.3%)	22 (21.2%)	129 (47.1%)			
Positive	124 (53.7%)	21 (48.8%)	145 (52.9%)	0.559		
Inflammatory BC						
No	200 (85.8%)	35 (85.4%)	235 (85.8%)			
Yes	33 (14.2%)	6 (14.6%)	39 (14.2%)	0.936		
Histological grade				0.228		
1						
2	112 (48.3%)	16 (38.1%)	128 (46.7%)			
3	120 (51.7%)	26 (61.9%)	146 (53.3%)			
Histologic type						
Ductal	198 (82.5%)	37 (86.0%)	235 (83.9%)			
Lobular	23 (9.6%)	4 (9.3%)	27 (9.6%)			
Others	16 (7.9%)	2 (4.7%)	18 (6.5%)	1.000		
HR						
Negative	86 (36.3%)	26 (60.5%)	112 (40.0%)			
Positive	151 (63.7%)	17 (39.5%)	168 (60.0%)	0.003		
Molecular subtype						
RH+HER2₋	112 (47.3%)	15 (34.5%)	127 (45.5%)		1	
RH+HER2+	39 (16.5%)	2 (4.8%)	41 (14.6%)		0.21 (0.05–0.93)	0.039
RH₋HER2+	28 (11.7%)	12 (27.9%)	40 (14.3%)		2.18 (0.93–5.11)	0.072
RH₋HER2₋	58 (24.5%)	14 (32.8%)	72 (25.6%)	0.007	1.09 (0.51–2.34)	0.826
Platelet count						
<264 G/L	112 (47.3%)	29 (67.4%)	141 (50.4%)		1	
≥264 G/L	125 (52.7%)	14 (32.6%)	139 (49.6%)	0.014	0.40 (0.21–0.75)	0.004
White blood cell count						
<6.75 G/L	109 (46.0%)	31 (72.1%)	140 (50.0%)		1	
≥6.75 G/L	128 (54.0%)	12 (27.9%)	140 (50.0%)	0.002	0.35 (0.17–0.72)	0.004
Neutrophil count						
<1.5 G/L	0 (0.0%)	1 (2.3%)	1 (0.4%)			
≥1.5 G/L	237 (100.0%)	42 (97.7%)	279 (99.6%)	0.154		
Lymphocyte count						
<1 G/L	7 (3.0%)	1 (2.3%)	8 (2.9%)	1.000		
≥1 G/L	230 (97.0%)	42 (97.7%)	272 (97.1%)			
NLR						
<2	106 (44.7%)	31 (72.1%)	137 (48.9%)		1	
≥2	131 (55.3%)	12 (27.9%)	143 (51.1%)	0.001	0.28 (0.13–0.58)	0.001
PLR						
<150	148 (62.4%)	29 (67.4%)	177 (63.2%)			
≥150	89 (37.6%)	14 (32.6%)	103 (36.8%)	0.532		

cT: clinical tumor size, cN: clinical node involvement, BC: breast cancer, HR: hormone receptor, NLR: neutrophil to lymphocyte ratio, PLR: platelet to lymphocyte ratio.

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
