# Peer review of "Inflammatory Blood Markers as Prognostic and Predictive Factors in Early Breast Cancer Patients Receiving Neoadjuvant Chemotherapy"

_cancers, 2020, doi:10.3390/cancers12092666_

Round 1

Reviewer 1 Report

This well-written manuscript by Corbeau and co-authors describes the platelet to lymphocyte ratio as an independent prognostic factor for survival in early breast cancer (EBC) patients. The basis of the study were clinical data of 280 EBC patients who have received neoadjuvant chemotherapy.

With 280 patients it is one of the biggest studies exanimating inflammatory blood markers (IBM) in EBC. Similar studies for EBM were published in the past, but most of them were dealing with Asian-based population. Contrary this current paper is working with a large population of western-patients. This demographic variable might be very important in terms of treatment side effects like toxicity and therefore differences in the IBM.

Comments:

  1. Since the paper includes many abbreviations, please check carefully if they are introduced correctly.
  2. Remove the second “patients” (P 2.2, first sentence)
  3. The conclusion section is missing
  4. Ethical statement (diary number?)
  5. Check carefully the references

Author Response

Please find below my response to this reviewer in bold. 

Comments:

  1. Since the paper includes many abbreviations, please check carefully if they are introduced correctly. The number of abbreviations has been decreased for a better comprehension.
  2. Remove the second “patients” (P 2.2, first sentence) Done
  3. The conclusion section is missing A conclusion section has been added
  4. Ethical statement (diary number?) Ethical statement has been added in the material and method section. 
  5. Check carefully the references Done

Reviewer 2 Report

Minor Concerns

  1. This content of this manuscript is filled with abbreviated terms; this reviewer recognizes that the authors have explained the abbreviated terms in the text of the manuscript before their extensive applications; however, due to the considerable numbers of abbreviated terms, it becomes somewhat difficult and to some degrees annoying; please reduce the number of abbreviations; in general, the recommended number of abbreviations is no more than three.

    2. Throughout the manuscript, the authors have erroneously used the term Her2; please correct this term to HER2.

    3. Please be consistent with presentation of % and the corresponding n values. For example, in page 2, lines 77 and 78, the authors have failed to provide the n value for “Seventy present of the patients……” and “Ninety-seven percent….” While, the value of n after a % is reported in the other parts of the manuscript (page 3, line 96)

    4. Page 7 of 15, lines 160-163. This sentence is exceptionally long and therefore, difficult to understand; please break down this sentence into two, preferably three sentences.

    5. Page 9 lines 196 and 236. The authors have used the term “correlation” erroneously. “The line should state “In addition, , as previously reported we found a highly significant correlation between PCR and survival…..” In the Statistic section, the authors have used statistical techniques which measures association not correlation. Please correct this error.

    6. Page 2 of 15, line 44, the authors have stated Breast cancer is the most common feminine cancer. Please change the term feminine to female.

    7. Page 2 of 15, line 45. Please change the term “Although..” to “Due..”

    8. Page 2 of 15, line 61, please change “We evaluated if there was an correlation” to “We evaluated if there were an association…”

    Major Concern :Inflammatory blood markers could be a response to diseases other than breast cancer. The authors have failed to provide any information about the co-existing adverse health conditions of the women who contributed to their study. Given their dataset and their patient population, it should be relatively
    easy for the authors to glean the information about co-morbid conditions in their patient population. These co-morbid health conditions should be included in their statistical analysis to ascertain removal of any bias in their statistical evaluation. Also, it would be quite valuable, if data on body mass index, proxy for obesity, are shared with the readers; furthermore, body mass index should be included in the statistical evaluation of predictive value of inflammatory blood markers as prognostic factor for survival and predictive factor for pathological complete response. 2. Section 5. Conclusions is missing. Thank you for giving me the opportunity to review this manuscript.

Author Response

Please find below, in bold the responses to the reviewer's comments. 

1.This content of this manuscript is filled with abbreviated terms (...) the number of abbreviation has been decreased to three so that it is easier to read, as asked.

2. Throughout the manuscript, the authors have erroneously used the term Her2; please correct this term to HER2. Done

3. Please be consistent with presentation of % and the corresponding n values. For example, in page 2, lines 77 and 78, the authors have failed to provide the n value for “Seventy present of the patients……” and “Ninety-seven percent….” While, the value of n after a % is reported in the other parts of the manuscript (page 3, line 96) The text has been completed as asked.

4. Page 7 of 15, lines 160-163. This sentence is exceptionally long and therefore, difficult to understand; please break down this sentence into two, preferably three sentences. Done : " Molecular subgroup was predictive for febrile neutropenia since 34.9% of positive hormone receptor-HER2 negative patients, 32.6% of negative hormone receptor-HER2 negative patients and 27.9% of negative hormone receptor-HER2 positive patients developed febrile neutropenia. Only 4.7% of patients with positive hormone receptor and positive HER2 status developed this toxicity (p= 0.007). "

5. Page 9 lines 196 and 236. The authors have used the term “correlation” erroneously. Done

6. Page 2 of 15, line 44, the authors have stated Breast cancer is the most common feminine cancer. Please change the term feminine to female. Done

7. Page 2 of 15, line 45. Please change the term “Although..” to “Due..” Disagree with the correction, the meaning of the sentence would change. We would prefer to keep the term "although".

8. Page 2 of 15, line 61, please change “We evaluated if there was an correlation” to “We evaluated if there were an association…” Done

Major Concern :(...) These co-morbid health conditions should be included in their statistical analysis to ascertain removal of any bias in their statistical evaluation. Since all our patients are receiving neo adjuvant chemotherapy there was no major comorbidity. This information has been added in the "Results" section. Also, it would be quite valuable, if data on body mass index, proxy for obesity, are shared with the readers; furthermore, body mass index should be included in the statistical evaluation of predictive value of inflammatory blood markers as prognostic factor for survival and predictive factor for pathological complete response. Unfortunately, the BMI was not accessible for the whole population. The impact of BMI can not be included in the statistical analysis. 

Section 5. Conclusions is missing. We have added a conclusion section.